# THE IMPACT OF TASK STRUCTURE, REPRESENTATIONAL GEOMETRY, AND LEARNING MECHANISM ON COMPOSITIONAL GENERALIZATION

**Samuel Lippl**
Center for Theoretical Neuroscience
Columbia University
New York, NY, USA
`samuel.lippl@columbia.edu`

**Kimberly Stachenfeld**
Google DeepMind and
Center for Theoretical Neuroscience
Columbia University
New York, NY, USA
`stachenfeld@deepmind.com`

## ABSTRACT

Compositional generalization (the ability to respond correctly to novel arrangements of familiar components) is thought to be a cornerstone of intelligent behavior. However, a theory of how and why models generalize compositionally across diverse tasks remains lacking. To make progress on this topic, we consider compositional generalization for kernel models with fixed, potentially nonlinear representations and a trained linear readout. We prove that they are limited to conjunction-wise additive compositional computations, and identify compositionality failure modes that arise from the data distribution and the model structure. For models in the representation learning (or "rich") regime, we show that networks *can* generalize on an important non-additive task (transitive equivalence) and give a mechanistic account for why. Finally, we validate our theory empirically, showing that it captures the behavior of a convolutional network trained on a set of compositional tasks. Taken together, our theory characterizes the principles giving rise to compositional generalization in models with fixed representations, shows how representation learning can overcome their limitations, and provides a taxonomy of compositional tasks that may be useful beyond the models considered here.

## 1 INTRODUCTION

Humans' understanding of the world is inherently compositional: once familiar with the concepts "pink" and "elephant," we can immediately imagine what a pink elephant looks like. Stitching together different concepts in this way allows humans to generalize far beyond our prior experience, preparing us to cope with unfamiliar situations and imagine things that do not yet exist (Frankland & Greene, 2020). The question of how to endow machine learning models with compositional abilities is a long-standing and historically vexing problem (Fodor & Pylyshyn, 1988; Lake et al., 2017; Battaglia et al., 2018; Lake & Baroni, 2018; Hupkes et al., 2020; Keysers et al., 2020). Recent breakthroughs (especially in language and computer vision) have led to massive improvements in models' compositional capacities, but in some cases, these models still fail spectacularly (Srivastava et al., 2023; Lewis et al., 2023; West, 2023; Ma et al., 2023; Chen et al., 2023). More generally, understanding the complex relationship between training data, model structure, and compositional generalization is still an unsolved problem, and it is often difficult to predict when networks will fail.

To make progress on this question, researchers often evaluate model performance on challenging benchmarks of compositional generalization (Johnson et al., 2017; Bahdanau et al., 2019; Ruis et al., 2020; Hupkes et al., 2020) or formally characterize model behavior on specific compositional tasks, clarifying the mechanisms giving rise to successful or unsuccessful generalization (Abbe et al., 2023; Jarvis et al., 2023; Lippl et al., 2024). However, it is unclear what the ability of a model to generalize compositionally on a specific task tells us about its compositional abilities in general, as the relationship among compositional tasks remains unclear.

The primary goal of this paper is to gain insight into how a model's representation and learning mechanism impact its compositional generalization capabilities. To this end, we leverage the lazy/rich dichotomy characterizing the two learning regimes of deep neural networks (Jacot et al., 2018; Chizat et al., 2019). In the lazy (or kernel) regime, neural networks behave like a kernel model with a fixed representation, for which we can formally relate representations and generalization capabilities. This allows us to derive key failure modes of compositional generalization. In the rich regime, neural networks learn data-dependent representations, which we show can overcome the compositional limitations imposed by the lazy regime.

Our specific contributions are as follows:

- We formulate a general task space for compositional generalization and characterize the full range of compositional motifs implemented by kernel models ("conjunction-wise additivity").
- This allows us to categorize tasks according to whether kernel models can solve them, which we show is a fundamental distinction not captured by existing criteria.
- For tasks solvable by kernel models, we highlight two failure modes ("memorization leak" and "shortcut distortion") and showcase their impact on two important compositional tasks.
- We then show how rich networks can overcome the limitations of conjunction-wise additivity.
- Finally, we validate our theory empirically by successfully capturing the behavior of deep neural networks.

Because our theory applies to a broad range of different tasks, our insights do not just clarify model behavior on specific compositional tasks but also shed light on their compositional abilities more generally. We see this work as a step toward building a more comprehensive theory of compositional generalization, allowing us to better predict and optimize the out-of-distribution behavior of models.

## 2 RELATED WORK

**Compositionality and modularity.** Compositionality is an important theme across human and machine reasoning problems, including visual reasoning (Lake et al., 2015; Johnson et al., 2017; Marino et al., 2019), language production (Hupkes et al., 2020), image generation (Okawa et al., 2023), and robotics (Zhou et al., 2022). Attempts to improve compositional generalization often leverage modular architectures, in the hopes that different modules will specialize for different components (Andreas et al., 2017). However, end-to-end training of these architectures often does not result in the desired modular specialization (Bahdanau et al., 2019; Mittal et al., 2022; Jarvis et al., 2023). Indeed, the more challenging problem than discovering modules might be learning how to use them, as even standard network architectures can develop specialized modules (Csordás et al., 2021; Hod et al., 2022). Notably, meta-learning can help networks use different modules correctly Schug et al. (2024). Indeed, even standard network architectures can develop specialized modules (Lepori et al., 2023), suggesting that the more challenging problem is how to correctly use them (Csordás et al., 2021; Hod et al., 2022). Pretraining large models on massive datasets may improve compositional generalization (Herzig et al., 2021; Furrer et al., 2021), but whether these models generalize successfully varies from task to task (Srivastava et al., 2023; Lewis et al., 2023; West, 2023). Our results characterize the specific modular computations arising from kernel models and relate failures to use them correctly to the task structure and inductive bias of the learning mechanism.

**Additive compositions and conjunctions.** Constraining a network to be additive (or implement another specific compositional function) can be a sufficient condition for compositional generalization (Lachapelle et al., 2023; Wiedemer et al., 2023). Our results reveal that kernel models are constrained to a conjunction-wise additive computation — this may be related to prior findings that language models encode many semantic concepts in an additive manner (Mikolov et al., 2013; Naito et al., 2021). Conjunctive codes (representations specific to a particular combination of features), on the other hand, have a long history in neuroscience (Alvarado & Rudy, 1992; Baker et al., 2002), and are theoretically linked to forming highly specific, episodic memories. Our results suggest that a mixture of conjunctive and compositionally additive codes may naturally arise from basic learning mechanisms. The resulting failure modes are related to prior work on shortcut learning and memorization (Hermann et al., 2023; Maini et al., 2023).

**Kernel and rich regime.** Prior work has revealed two key strategies by which deep neural networks learn (Chizat et al., 2019; Woodworth et al., 2020). In the kernel (or "lazy") regime (which can be induced with large initial weights or wide networks), the networks' learning is well approximated by gradient descent on a model with a fixed representation (Jacot et al., 2018). In the rich regime (brought forth, for example, by small initial weights, small width, or loss functions like cross-entropy), the networks learn structured (i.e. abstract and sparse) representations over the course of learning (Savarese et al., 2019; Chizat & Bach, 2020; Lyu & Li, 2020; Saxe et al., 2022). Notably, the rich regime gives rise to better generalization and more human-like behavior in neural networks (Fort et al., 2020; Vyas et al., 2022; Flesch et al., 2022).

**Kernel trick and norm minimization.** When using gradient descent (or similar learning algorithms) to train the readout weights of a model, it has long been known that this model depends on its representation $r(x)$ only through its induced kernel $K(x, x') = \langle r(x), r(x') \rangle$ (Schölkopf, 2000). Specifically, when trained on a dataset $\{(x_i, y_i)\}_{i=1}^n$, their function can be written in its "dual form":

$$f(x) = \sum_{i=1}^n a_i K(x, x_i). \tag{1}$$

Further, gradient descent selects among the models consistent with the training data that with the smallest $\ell_2$-norm on the readout weights ("norm minimization") (Soudry et al., 2018; Gunasekar et al., 2018). This is also true of neural networks in the lazy regime. We here leverage the constraints apparent from this dual form to characterize the set of possible compositional computations in kernel models and further analyze the specific kernels induced by different network architectures to predict generalization under norm minimization. This has been a broadly popular method in theoretical machine learning and neuroscience (see e.g. Canatar et al., 2021; 2023) and has also been applied to compositional tasks specifically (Lippl et al., 2024; Abbe et al., 2023).

**Comparison with prior theoretical work.** Lippl et al. (2024) analyze how kernel models generalize on a specific compositional task, transitive inference. Further, Abbe et al. (2023) analyze how kernel models generalize on tasks with binary components in the limit of infinite components. In contrast, this work analyzes tasks with more general compositional structure, derives exact constraints for finite numbers of components, and highlights failure modes that arise specifically for finite numbers of components. Finally, Jarvis et al. (2023) analyze how deep linear networks generalize on a family of tasks with a specific input-output relation. In contrast, in Section 4, we characterize constraints on the generalization behavior of kernel models on tasks with arbitrary input-output relations.

## 3 METHODS

### 3.1 A GENERAL COMPOSITIONAL TASK SPACE

We assume that each task input $I$ has a fixed number of distinct components: $I = (I_c)_{c=1}^C$, where each $I_c$ is drawn from some set of possible component values $\mathcal{I}_c$. We consider a set of $n$ training pairs $\mathcal{D}^{(\text{train})} = \{(I^{(i)}, y^{(i)})\}_{i=1}^n$, $y^{(i)} \in \mathbb{R}$, analyzing generalization on the remaining $m$ test set pairs $\mathcal{D}^{(\text{test})} = \{(I^{(i)}, y^{(i)})\}_{i=n+1}^{n+m}$. This captures a broad range of compositional tasks, for example:

**Addition.** This task consists of two components $(I_1, I_2)$ with unobserved assigned values $v_1(I_1)$ and $v_2(I_2)$. The target is the sum of those values: $y = v_1(I_1) + v_2(I_2)$. If each item has been seen in combination with at least one other item, a model with an additive structure could generalize to novel combinations of items. We consider nine input elements $[-4], [-3], \ldots, [4]$, with associated values $-4, -3, \ldots, 4$ and consider four different training and test splits (Fig. 1a): (i) an extrapolation task (train only on pairs with at least one $[0]$), (ii) an asymmetric extrapolation task (train only on pairs including one $[-4]$), (iii) an interpolation task (training set contains all trials with a $[-4]$ or $[4]$), and (iv) a dispersed task (each item occurs in exactly two randomly sampled training pairs).

**Transitive equivalence.** Given an unobserved (and arbitrary) equivalence relation, the task is to determine whether two presented items $(I_1, I_2)$ are equivalent. The model should generalize to novel item pairs using transitivity ($A = B$ and $B = C$ imply $A = C$) (Fig. 1b).

**Context-dependent decision-making (CDM).** This task has three input components $(\text{co}, \text{feat}_1, \text{feat}_2)$. The context (co) has two possible values specifying whether $\text{feat}_1$ or $\text{feat}_2$ determine the response. Both features have six possible values which are split up in two categories (Fig. 1c). If the model

Figure 1: Compositional tasks. **a**, Different training sets for symbolic addition. The grid represents nine components with associated values $-4, -3, \ldots, 4$ and each tile represents a data point in the training set. **b**, Transitive equivalence. **c**, Context-dependent decision-making: in context 1, feat. 1 determines the output; in context 2, feat. 2 determines the output. **d**, For the test set, different subsets of the bottom right orthant are held-out in both contexts (for CDM-3, the entire orthant is held-out; for CDM-1/2, different subsets are held-out). **e**, Invariance and partial exposure. Translucent tiles indicate the generalization set.

has learned this context dependence, it should be able to generalize to novel combinations of items. We evaluate on the orthant of item combinations for which $\text{feat}_1$ indicates Cat. 2 and $\text{feat}_2$ indicates Cat. 1 (Fig. 1d). In the most extreme generalization test (*CDM-3*), we leave out the entire orthant, in easier versions we leave out conjunctions of two or one of those features (*CDM-2* and *CDM-1*).

**Invariance.** The model sees two items $(I_1, I_2)$ and its response should only depend on $I_2$. In the training set, $I_1$ has a constant value, which is changed in the test set (Fig. 1e). Compositional generalization is possible if the model ignores features that have not varied.

**Partial exposure.** Invariance, with an added training trial showing a new value for $I_1$ (Dasgupta et al., 2022). This gives $I_1$ a spurious correlation with the target that may distort generalization.

## 3.2 MODELS

We consider three types of models. First, we consider **kernel models** optimized with norm minimization. We either explicitly specify their kernel $K(x, x')$ or consider a fixed representation $r(x)$ which arises within a neural network receiving as input a concatenation of onehot vectors indicating the identity of each component. $K$ (resp. $r$) is fixed, whereas the linear readout is learned. Second, we train **ReLU neural networks** with a single hidden layer through backpropagation, using initial weight distributions with different variances to put them into the lazy or rich regime (Chizat et al., 2019). Finally, we consider a **convolutional neural network** with four convolutional layers and two fully connected layers.

## 3.3 COMPOSITIONAL STRUCTURE

The trial-by-trial similarity between different inputs can depend on two factors. First, they may overlap on a specific set of components. We denote this overlap by

$$O(I, I') := \{c \in \{1, \ldots, C\} : I_c = I'_c\}. \tag{2}$$

Second, their components may have additional similarity structure. For example, certain MNIST digits look more similar to each other. Such a representation could solve a compositional task in a non-compositional manner. Suppose, for example, that a model, for whatever reason, represents equivalent items in the transitive equivalence task as more similar. This would enable the model to generalize successfully simply by responding to $(A, B)$ and $(A, C)$ similarly, but does not actually depend on the compositional structure of the task. Because we are interested in compositional generalization, we assume that model representations only reflect the inputs' compositional structure:

**Definition 3.1.** A kernel $K$ is "compositionally structured" if it only depends on its items' overlap, i.e. if for any set of compositional inputs $I, I', J, J' \in \mathcal{I}$, $O(I, I') = O(J, J')$, then $K(I, I') = K(J, J')$.

Note that if $r(x)$ arises from a random deep neural network with inputs concatenating onehot vectors for each component, it is compositionally structured in expectation. This is because this input is compositionally structured (as the similarity between two inputs is equal to their number of overlaps) and the output kernel of densely connected neural networks with random weights only depends on their input kernel (i.e. for a network $\phi$, if $K(x_1, x_1') = K(x_2, x_2')$, then $K(\phi(x_1), \phi(x_1')) = K(\phi(x_2), \phi(x_2')))$.

# 4 LAZY MODELS IMPLEMENT CONJUNCTION-WISE ADDITIVE COMPUTATIONS

## 4.1 TASKS WITH TWO COMPONENTS

On tasks with two components, the dual form (Eq. 1) immediately reveals a fundamental constraint.

**Proposition 4.1.** *If a compositionally structured kernel model $f$ has been trained on data $\mathcal{D}^{(train)}$ from a compositional task with $C = 2$ components, there exist functions $f_1 : \mathcal{I}_1 \to \mathbb{R}$, $f_2 : \mathcal{I}_2 \to \mathbb{R}$, and $f_{12} : \mathcal{I} \to \mathbb{R}$ s.t.*

$$f(I) = \begin{cases} f_1(I_1) + f_2(I_2) + f_{12}(I_1, I_2) & \text{if } I \in \mathcal{D}^{(train)}, \\ f_1(I_1) + f_2(I_2) & \text{else.} \end{cases}$$

*Proof.* See Appendix A. □

Intuitively, compositionally structured representations can consist of three different populations: one coding for the first component by itself, one coding for the second component by itself, and one coding for their unique conjunction (Fig. 2a). Linear readout learning corresponds to assigning a value to each of those components and adding those values together (Proposition 4.1). When exposed to novel combinations of items, the specific conjunction of items has never been seen before (i.e. $f_{12}(I) = 0$) and so the model can only rely on the additive computation (i.e. $f_1(I_1) + f_2(I_2)$).

Proposition 4.1 is an impossibility theorem. While many kernel models can learn arbitrary functions (Cybenko, 1989), our result shows that their compositional generalization is constrained to summing up values associated with each component. This directly implies that compositionally structured kernel models cannot generalize on certain tasks. In particular, transitive equivalence requires a non-additive (*XOR*-type) equality operation. As a result, while kernel models can memorize the training data, they provably cannot generalize to the test data. Importantly, this constraint does not depend on whether the conjunction $f_{12}(I_1, I_2)$ is used for the training set, but directly arises from the functional constraint on the test set.

## 4.2 TASKS WITH MORE THAN TWO COMPONENTS

For more than two components, we have a combinatorial explosion of conjunctions to contend with, as terms in the model can depend on any combination of components (e.g. $f_1(I_1), f_{12}(I_1, I_2), f_{123}(I_1, I_2, I_3)$). We denote each combination (e.g. 12) as $S \in \mathcal{S}$, and let $\mathcal{S}$ be the set of all combinations $S \subseteq \{1, \ldots, C\}$. Each $S \in \mathcal{S}$ indexes input components $(I_i)_{i \in S}$, which we denote $I_S$ for convenience. The model assigns a value of $f_S(I_S)$ to each combination, then sums these values (Fig. 2b).

**Proposition 4.2.** *Consider a compositionally structured kernel model $f$ that has been trained on a compositional task with input space $\mathcal{I}$ and training set $\mathcal{D}^{(train)}$. For an input $I \in \mathcal{I}$, we define the set of overlaps*

$$\mathcal{O}(I) := \{S \in \mathcal{S} | \exists_{I' \in \mathcal{D}^{(train)}} \text{ s.t. } S \subseteq O(I, I')\}.$$

*Then, there exists a set of functions $(f_S)_S$ s.t. for all $I$, $f(I) = \sum_{S \in \mathcal{O}(I)} f_S(I_S)$.*

*Proof.* See Appendix A. □

Just like in Proposition 4.1 (a special case), behavior here depends on the relationship between $I$ and $\mathcal{D}^{(\text{train})}$: for every possible conjunction $S$, the model can assign a specific value $f_S(I_S)$ to $I_S$ only if it has seen $I_S$ during training.

We call tasks that can be solved in this manner "conjunction-wise additive," as opposed to "component-wise additive." For example, context-dependent decision-making can be implemented with a conjunction between context and feature (see Appendix D.2). As these conjunctions are all observed during training, this task is conjunction-wise additive.

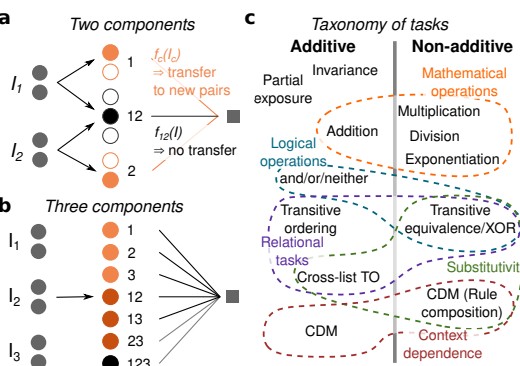

### 4.3 A Compositional Task Taxonomy

Conjunction-wise additivity is a natural way of carving up the space of compositional tasks: for additive tasks we have a candidate compositional mechanism and can characterize the conditions on task structure and representational geometry that yield successful generalization. We do so in the next section. For non-additive tasks, we know that kernel models are fundamentally unable to perform them successfully and that we should look for alternative learning models. We do so in Section 6. Our criterion cuts across a range of other compositional task categories (Fig. 2c, Appendix D), highlighting, for example, a fundamental difference between applying the transitive rule to ordered relations (which are additive; Lippl et al. 2024) and equivalence relations (which are non-additive).

Figure 2: Kernel models implement conjunction-wise additive computations. **a**, Any compositionally structured representation is a mixture of a compositional and a conjunctive representation. Both have a distinct impact on test trial behavior. **b**, Compositionally structured representations have a different subpopulation for each conjunction of components and therefore implement a conjunction-wise additive computation. Each conjunction generalizes to test trials containing the same conjunction. **c**, Conjunction-wise additivity separates tasks into those that can be solved by lazy models and those that cannot.

## 5 The Impact of Representational Geometry and Task Structure

We now investigate how generalization depends on learning mechanism, model, and task structure.

### 5.1 Network Architecture and Nonlinearities Impact Representational Geometry on Compositional Tasks

We determine how neural networks with random Gaussian weights represent compositional inputs (presented as one-hot vectors) as a function of their depth and nonlinearity (Appendix B). As noted, these networks give rise to compositionally structured representations. In fact, the resulting kernel $K(I, I')$ only depends on how many components $I$ and $I'$ have in common, rather than their specific overlap.

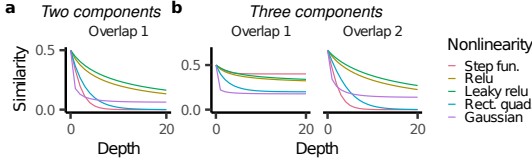

Figure 3: Normalized representational similarity between overlapping items for **a**, two components and **b**, three components.

This means that for two components, this kernel has different similarities for trials overlapping with zero ($\kappa_0$), one ($\kappa_1$), or two (i.e. all; $\kappa_2$) components. Below we find that the "normalized" magnitude of $\kappa_1$, $\hat{\kappa}_1 := \frac{\kappa_1 - \kappa_0}{\kappa_2 - \kappa_0}$, is particulary impactful on behavior. The one-hot input has $\hat{\kappa}_1 = 0.5$ (indicating a fully compositional representation). As the network becomes deeper, $\hat{\kappa}_1$ decreases, indicating a more conjunctive representation (as overlapping trials are less similar, their conjunction is emphasized more strongly).

For tasks with three components, there are different similarities $\kappa_p$ for trials overlapping with $p = 0, 1, 2, 3$ components. We consider $\hat{\kappa}_2 := \frac{\kappa_2 - \kappa_0}{\kappa_3 - \kappa_0}$, which describes whether the representation

emphasizes conjunctions of two components or conjunctions of three components, and $\hat{\kappa}_1 := \frac{\kappa_1 - \kappa_0}{\kappa_2 - \kappa_0}$, which describes whether the representation emphasizes single components or conjunctions of two components. Generally, both quantities decrease with increasing depth, but the nonlinearity affects their relative trajectory.

## 5.2 MEMORIZATION LEAK AND SHORTCUT DISTORTION

Generally speaking, norm minimization tends to learn distributed weights. For compositional tasks, it therefore tends to rely on many different conjunctions. This gives rise to two important failure modes:

**Memorization leak.** First, the model tends to use the conjunctive population even when this is not necessary to solve the task. In a purely additive task, this necessarily distorts the model's additive structure (Proposition 4.1), impairing its compositional generalization. Further, the more conjunctive the representation, the worse the distortion. We call this effect the "memorization leak" (see Fig. 12b). To see it in action, we analytically solve model generalization for component invariance (Appendix D.3), finding that its test margin is expressed as $m = \frac{\hat{\kappa}_1}{1 - \hat{\kappa}_1}$ (see Fig. 12a). This means that for a fully compositional representation ($\hat{\kappa}_1 = 0.5$), its training and test margins are both one. However, as $\hat{\kappa}_1$ decreases, the model increasingly memorizes the training set, resulting in a decreased margin on the test set (while the training set margin remains constant).

**Shortcut distortion.** Many compositional tasks have training set imbalances. Consider the partial exposure task. If the model used item 1 to solve the task, it would get two out of three training examples correct and could memorize the last data point. This is an example of a "shortcut" (Geirhos et al., 2020), which only works on (part of the) training set and which generalizes incorrectly (see Fig. 12c). Norm minimization ends up partially relying on this strategy as this decreases the $\ell_2$-norm of the readout weights. As a result, the test margin for the partial exposure task decreases even more strongly as a function of $\hat{\kappa}_1$: $m = \frac{2\hat{\kappa}_1^2}{1 - 2\hat{\kappa}_1^2}$ (Fig. 12a; Appendix D.4).

We now consider two important additive tasks: addition and context-dependent decision-making.

## 5.3 SYMBOLIC ADDITION

For symbolic addition, we solve model behavior on the extrapolation task analytically (Appendix D.1). We find that the model underestimates the items' values: $f_c(v) = 2\hat{\kappa}_1 v$ (Fig. 4a). This is due to a memorization leak and becomes worse for smaller $\hat{\kappa}_1$.

To investigate the impact of different training sets on the memorization leak, we determine the inferred values for a partially conjunctive representation ($\kappa_0 = 0$, $\kappa_1 = 0.4$, $\kappa_2 = 1$, Fig. 4b; see Appendix D.1 for more detailed simulations). We find that asymmetric extrapolation again yields compressed values and is

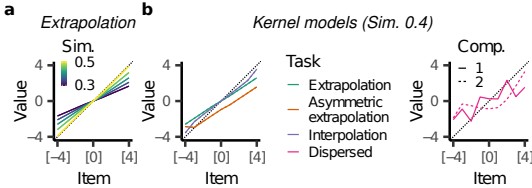

Figure 4: Kernel models trained on symbolic addition are affected by a memorization leak. Inferred values for **a**, the extrapolation task as a function of $\hat{\kappa}_1$ and **b**, the different training sets ($\hat{\kappa}_1 = 0.4$).

even more challenging than extrapolation. While the interpolation task is easier (as expected), the model, perhaps surprisingly, still suffers from a memorization leak, underestimating intermediate values. Finally, the dispersed training set yields behaviors somewhere in between the extreme cases of asymmetric extrapolation and interpolation. In that case, the model may also infer different values for the two components, as the training set is no longer symmetric (and the model has no notion of shared values between the two components).

## 5.4 CONTEXT-DEPENDENT DECISION-MAKING

We now study context-dependent decision-making. This task represents a bigger challenge to kernel models, as they must identify the correct context-feature conjunctions out of the many possible conjunctions available to them. We determine whether the model generalizes correctly as a function

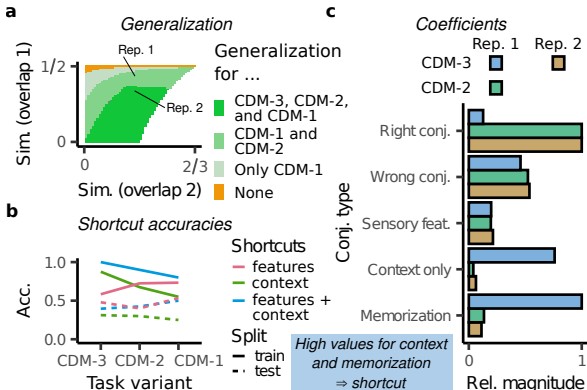

Figure 5: Shortcut distortion explains model failure on context-dependent decision-making. **a**, Generalization performance as a function of representational geometry and task variant. **b**, Possible shortcuts and their accuracy. **c**, We group the conjunctive coefficients into different categories (see Appendix D.2) and compute their average magnitude. For each model, we normalize these magnitudes so the maximal magnitude is one. The failure for Rep. 1 to generalize on CDM-3 is explained by a shortcut indicated by large coefficients associated with context and memorization.

of representational geometry and training set. We find that for *CDM-3*, successful generalization in kernel models strongly depends on the representational geometry (Fig. 5a). For *CDM-2* and *CDM-1*, generalization is much less dependent on the specific representational geometry.

To unpack these behaviors in more detail, we note that there are multiple shortcuts in the training data (Fig. 5b). In particular, context by itself can classify a substantial proportion of the training data correctly and so can the features without any context. However, these shortcuts yield generalization behavior that is at best at chance level.

We now compare two representations: one that is unsuccessful on *CDM-3* but successful on *CDM-2* (Rep. 1) and one that is successful on both task variants (Rep. 2). Looking at the models' conjunctive coefficients, we determine their average magnitude within different categories (see Appendix D.2 for details) (Fig. 5c). Indeed, the two successful models both attribute the largest magnitude to the correct conjunction between context and features. In contrast, the unsuccessful model instead relies on a mixture of the context component and memorization, indicating that it inferred a shortcut solution.

We can explain this failure in terms of the model's representation (i.e. Rep. 1), which represents individual components more prominently than conjunctions of two components (at least compared to Rep. 2). As a result, using a mixture of context (an individual component) and memorization is more norm-efficient than a conjunction of two components. For *CDM-2*, the context shortcut is much less useful (Fig. 5b) and, as a result, the model with Rep. 1 also learns to rely on the correct conjunctions.

In sum, we have here described the substantive challenges that lie in learning a conjunction-wise additive function and illustrated the use of kernel models to understand how these challenges are exacerbated or alleviated by different task structures and representational geometries.

## 6 RICH NETWORKS CAN OVERCOME THE LIMITATIONS OF CONJUNCTION-WISE ADDITIVITY

We now discuss how other learning mechanisms can overcome the limitations of conjunction-wise additivity. Specifically, we compare ReLU networks trained on transitive equivalence (a non-additive task) in the lazy and rich regime (Fig. 6a). In the lazy regime, they are approximated by a kernel model and thus do not generalize. In contrast, neural networks in the rich regime generalize correctly[1] (Fig. 6a).

---

[1]Cross-entropy eventually yields representation learning for any initial magnitude (Lyu & Li, 2020). However, this may require much longer training than investigated here (Kumar et al., 2023).

To explain why this is the case, we note that neural networks in the rich regime are biased to having weights with a low overall $\ell_2$-norm (Lyu & Li, 2020; Chizat & Bach, 2020; but see Vardi & Shamir, 2021). In particular, a ReLU network with one hidden layer is biased to learn a sparse set of hidden features (Savarese et al., 2019; Chizat & Bach, 2020). Transitive equivalence consists in multiple sets of equality relations (e.g. $A = B$ and $D = E$) and it is well known that ReLU networks learn such a nonlinear (*XOR*-type) problem by specializing one unit to each of the four conjunctions (Brutzkus & Globerson 2019; Saxe et al. 2022; Fig. 6b). Further, different sets of these equality relations have overlapping items (e.g. $A = B$ and $C = B$).

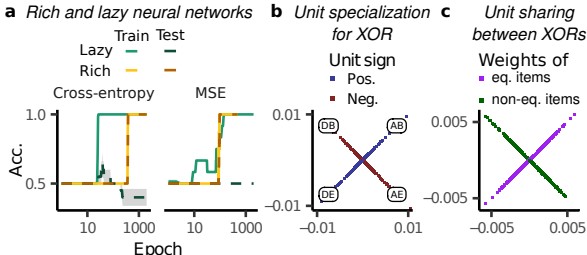

Figure 6: Rich neural networks can generalize on transitive equivalence. **a**, Generalization over time in the lazy and rich regime. **b**, The weights of the network in the subspace corresponding to one underlying *XOR*-task. **c**, Weights for the same unit are plotted against each other and colored by whether they correspond to equivalent items (purple) or non-equivalent items (green).

Because of their sparse inductive bias, ReLU networks use the same set of units for these overlapping conjunctions (e.g. $(A, B), (A, C)$, and $(B, C)$). This means that the same conjunction also generalizes to unseen item combinations (e.g. $(A, C)$), which gives rise to generalization. Importantly, our theoretical argument is corroborated by empirical simulations: we find that each hidden network unit has the exact same weights for equivalent items (Fig. 6c).

In sum, neural networks' known capacity for abstraction provides them with an additional compositional motif. Here we have illustrated this within one example task. As a notable contrast highlighted by our findings, transitive equivalence and transitive ordering, while involving the same rule, require fundamentally different networks to be solved. Note that on symbolic addition and context-dependent decision-making, rich neural networks exhibit largely additive behavior (see Appendix D). This suggests that the kernel theory can still be useful for understanding network behavior in the rich regime. Future work should investigate rich-regime compositional computations in more detail.

# 7 EXPERIMENTS IN DEEP NEURAL NETWORKS

So far we focused on toy domains with tabular representations and small MLPs, for their analytical tractability. To see whether our theory can help us understand different, larger-scale neural networks with more complex, interrelated inputs, we train convolutional neural networks on a version of the considered tasks involving concatenated MNIST digits (represented in different channels, e.g. with different colors). These tasks have exactly the same structure as the previous tasks, except that the one-hots denoting different numerical symbols are replaced by images of corresponding MNIST digits. Rather than being a single instance, each possible item is now one MNIST category. Notably, MNIST digits are *not* compositionally structured — for example, ones and sevens are more visually similar to each other than ones and twos. To control for the input similarity structure, we randomly permuted the assignment of MNIST digits to values for each ($n = 10$) experiment (e.g. the image for "6" might have value 1). Further, to compare the network behaviors to our theoretical predictions, we fit a conjunction-wise additive function to the network output using linear regression ("additivity analysis"; see Appendix C.2).

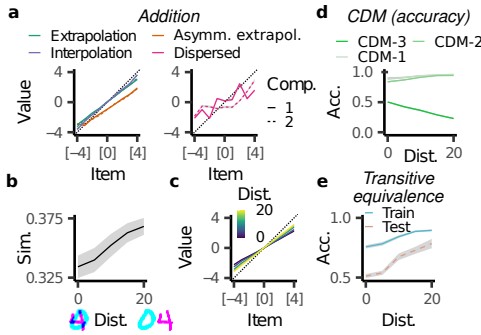

Figure 7: The kernel theory can explain convolutional networks trained on compositional tasks with MNIST digits. **a**, Inferred values on symbolic addition (see Fig. 4a). **b**, Rep. sim. in an intermediate layer for different distances between the digits. **c**, Inferred values on the extrapolation task for different distances. **d**, Accuracy on context-dependent decision-making (see Fig. 5). **e**, Accuracy on transitive equivalence.

First, we consider the different task variants of symbolic addition, training the network on a total of 20,000 randomly generated samples for 100 epochs and test its compositional generalization on new handwritten digits. Remarkably, we find that the kernel theory matches network behavior almost perfectly. First, the network's average prediction (across the different possible permutations of the digit categories) is well predicted by an additive structure (Fig. 10b). Further, the network's inferred values, across all tasks, are extremely similar to those in kernel models (Fig. 7a). This suggests that, at least in terms of its compositional generalization, the network is well described by the kernel theory.

To investigate the effect of increased conjunctivity, we varied the distance between the two digits (Fig. 7b), hypothesizing that if digits are closer, their representation should be more conjunctive. To test this, we determined the representational similarity in the network, normalizing so that two trials with identical digit categories had a similarity of one and two trials with distinct digit categories had a similarity of zero. We found that the similarity between overlapping digit categories was indeed smaller for lower distances, indicating a more conjunctive representation (Fig. 7b). Further, the values inferred on the extrapolation task are more compressed for smaller distances, confirming that more conjunctive inputs exacerbate the memorization leak in convolutional networks as well (Fig. 7c).

Next, we considered context-dependent decision-making. Once again we replace the onehots with different MNIST digits and train networks on a total of 30,000 samples for 100 epochs. Again, behavior was consistent with our kernel theory's predictions: the networks had better-than-chance accuracy on *CDM-1* and *CDM-2*, but not *CDM-3* (Fig. 7d). Further, the additivity analysis revealed that the networks trained on *CDM-3* relied on a context-based shortcut, though they learned the remaining training data by relying on sensory features rather than memorization (Fig. 11).

Finally, we considered transitive equivalence, training the network on 20,000 samples for 150 epochs. We found that if the digits were presented to the network in different channels but the same location, the network did not generalize compositionally at all. However, with increasing distance, the network started to improve its compositional generalization (Fig. 7e). This demonstrates that a convolutional network can benefit from non-additive compositional motifs.

## 8 CONCLUSION

This work is a step toward formalizing the relationship between model structure and compositional behaviors. We first described the full range of compositional computations ("conjunction-wise additive") that can be implemented by kernel models, distinguishing between additive tasks, which are solvable by kernel models, and non-additive tasks, which are not. For additive tasks, we highlighted important failure modes impacting generalization: memorization leak and shortcut distortion. For non-additive tasks, we showed how rich neural networks can overcome the limitations of conjunction-wise additivity. Finally, we validated our theory by showing that it captures behavior in deep neural networks. Future work could extend this analysis to other architectures (e.g. Transformers) and compositional tasks involving e.g. dynamic-length inputs and outputs and pretrained representations.

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

## A    PROOF OF PROPOSITION 4.2

**Proposition 4.2.** *Consider a compositionally structured kernel model $f$ that has been trained on a compositional task with input space $\mathcal{I}$ and training set $\mathcal{D}^{(train)}$. For an input $I \in \mathcal{I}$, we define the set of overlaps*

$$\mathcal{O}(I) := \left\{ S \in \mathcal{S} | \exists_{I' \in \mathcal{D}^{(train)}} \text{ s.t. } S \subseteq O(I, I') \right\}.$$

*Then, there exists a set of functions $(f_S)_S$ s.t. for all $I$, $f(I) = \sum_{S \in \mathcal{O}(I)} f_S(I_S)$.*

*Proof.* Because $K$ is compositionally structured, its similarity $K(I, I')$ only depends on the overlap $O(I, I') \subseteq \{1, \ldots, C\}$. We denote the similarity for trials overlapping in $S \subseteq \{1, \ldots, C\}$ by $\kappa_S$ and define the overlap in the training dataset as

$$\mathcal{D}_S(I) := \left\{ I' \in \mathcal{D}^{(\text{train})} | \forall_{c \in S} I_c = I'_c \right\}. \tag{3}$$

The key idea is to decompose $\mathcal{D}^{(\text{train})}$ into these different overlaps in order to separate the sum into its components. However, by our definition, the datasets $\mathcal{D}_S(I)$ are not disjoint. Indeed, $S \subseteq S'$ implies $\mathcal{D}_{S'}(I) \subseteq \mathcal{D}_S(I)$ and in particular $\mathcal{D}_\emptyset(I) = \mathcal{D}^{(\text{train})}$. To adjust for this, we define $\delta_S$ as the similarity added by $\kappa_S$ to the similarity between conjunctions with one component fewer, recursively defining

$$\delta_\emptyset = \kappa_\emptyset, \quad \delta_S = \kappa_S - \sum_{S' \subsetneq S} \delta_{S'}. \tag{4}$$

We then decompose

$$f(I) = \sum_{I' \in \mathcal{D}^{(\text{train})}} a_{I'} K(I, I') = \sum_{S \subseteq \{1, \ldots, C\}} \delta_S \sum_{I' \in \mathcal{D}_S(I)} a_{I'}. \tag{5}$$

This equality obtains because for each $I' \in \mathcal{D}^{(\text{train})}$,

$$\sum_{S : I' \in \mathcal{D}_S(I)} \delta_S = \delta_{O(I, I')} + \sum_{S \subsetneq O(I, I')} \delta_{S'} = \kappa_{O(I, I')} = K(I, I'), \tag{6}$$

which is true by definition. We note that for $S \notin \mathcal{O}(I), \mathcal{D}_S(I) = \emptyset$. Defining

$$f_S(I) := \delta_S \sum_{I' \in \mathcal{D}_S(I)} a_{I'}, \tag{7}$$

proves the proposition. $\square$

Note that Proposition 4.1 follows as a special case of Proposition 4.2.

## B    COMPUTING THE REPRESENTATIONAL SIMILARITIES

Note that all nonlinearities we consider are dot-product kernels, which only depend on the correlation between their inputs. For a given trial pair $(I, I')$, the input representation has a correlation

$$\frac{|O(I, I')|}{C},$$

as exactly those onehots which are overlapping are identical. This implies, in particular, that the similarity between network representations also only depends on the number of overlaps, $|O(I, I')|$. We then leverage the dot-product kernels, derived in previous work (Williams, 1996; Cho & Saul, 2009; Han et al., 2022), to compute the resulting similarities.

## C    DETAILED METHODS

### C.1    MODELS

**Kernel model.**    We fit the kernel methods by hand-specifying the kernel and fitting either a support vector regression or classification using `scikit-learn` (Pedregosa et al., 2011).

**Rich and lazy ReLU networks.** All networks were trained with Pytorch and Pytorch Lightning (Paszke et al., 2019). We consider ReLU networks with one hidden layer and $H = 1000$ units. We initialize by $\sigma\sqrt{2/H}$, considering $\sigma \in [10^{-6}, 1]$. In particular, when reporting results on rich networks (without further specification), we assume $\sigma = 10^{-6}$. When reporting results on lazy network, we assume $\sigma = 1$.

**Convolutional neural networks.** We consider networks with four convolutional layers (kernel size is five, two layers have 32 filters, two have 64 filters) and two densely connected layers (with 512 and 1024 units). Each layer is followed by a ReLU nonlinearity, and the convolutional stage is followed by a max pooling operation. All weights are initialized with He initialization (He et al., 2015).

## C.2 Additivity analysis

We analyze how well a conjunction-wise additive computation can describe network behavior. Specifically, we consider as the set of possible features a concatenation of one-hot vectors coding for each possible conjunction. We then remove all features that are constant at zero on the training dataset and use linear regression to try and predict network behavior on both training and test set for all remaining features. The resulting $R^2$ defines the "additivity" of the network behavior (i.e. $R^2 = 1$ indicates full conjunction-wise additivity). Furthermore, we can use the inferred values assigned to these different conjunctions to compare kernel models, rich and lazy networks, and convolutional networks. Note that for the convolutional networks, we first average the model predictions across all different images instantiating a given compositional input.

## D Analysis of compositional tasks

### D.1 Symbolic Addition

#### D.1.1 Theory

To get intuition for this problem, we first consider the case where $\mathcal{D}^{(\text{train})} = \{(I_1, I_2) | I_1 = J_1 \vee I_2 = J_2\}$ for two items $J_c \in \mathcal{I}_c$ with $v_c(J_c) = 0$. Further, we assume that the average values are also zero, i.e. $\sum_{I \in \mathcal{I}_c} v_c(I) = 0$ for $c = 1, 2$. Note that the extrapolation task represents an instance of such a task.

In this case, the dual problem is especially easy to solve. Specifically, there are three types of training data: $y_{11}$, $y_{1j}$ with $j > 1$ and $y_{i1}$ with $i > 1$ (we denote the $J$ item by 1). The corresponding dual equations are

$$0 = v_1(I_1) + v_2(I_1) = y_{11} = a_{11}\kappa_2 + \sum_{j=2}^{n} a_{1j}\kappa_1 + \sum_{i=2}^{m} a_{i1}\kappa_1, \tag{8}$$

$$v_2(I_l) = v_1(I_1) + v_2(I_l) = y_{1l} = a_{1l}\kappa_2 + \sum_{j=1, j\neq l}^{n} a_{1j}\kappa_1 + \sum_{i=2}^{m} a_{i1}\kappa_0, \tag{9}$$

$$v_1(I_k) = v_1(I_k) + v_2(I_1) = y_{k1} = a_{k1}\kappa_2 + \sum_{i=1, i\neq k}^{m} a_{i1}\kappa_1 + \sum_{j=2}^{n} a_{1j}\kappa_0. \tag{10}$$

We now denote averages over selected dual coefficients by

$$b_1 := \sum_{i=2}^{m} a_{i1}, \quad b_2 := \sum_{j=2}^{n} a_{1j}. \tag{11}$$

We further note that

$$\sum_{l=2}^{n} y_{1l} = \sum_{k=2}^{m} y_{k1} = 0, \tag{12}$$

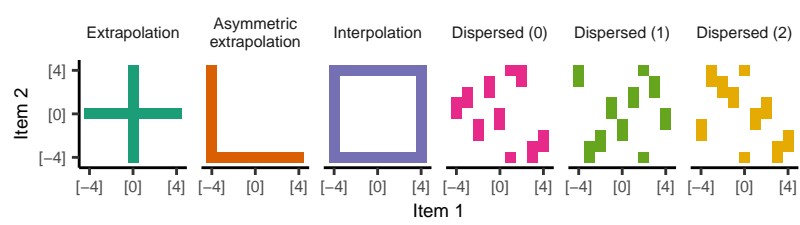

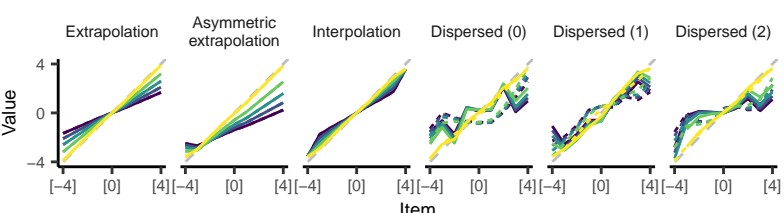

Figure 8: **a**, Training sets for addition. **b**, Inferred values for different training sets and kernels.

by a simple application of the assumption we made above. Rewriting (8) and summing over $l$ (and $k$) for (9) (and (10)) thus results in

$$0 = \kappa_2 a_{11} + \kappa_1 b_1 + \kappa_1 b_2, \tag{13}$$
$$0 = (n-1)\kappa_1 a_{11} + ((n-1)\kappa_1 + \kappa_2)b_1 + (m-1)\kappa_0 b_2, \tag{14}$$
$$0 = (m-1)\kappa_1 a_{11} + ((m-1)\kappa_1 + \kappa_2)b_2 + (n-1)\kappa_0 b_1. \tag{15}$$

This implies that $a_{11} = b_1 = b_2 = 0$. Thus,

$$v_2(I_l) = a_{1l}(\kappa_2 - \kappa_1), \quad v_1(I_k) = a_{k1}(\kappa_2 - \kappa_1). \tag{16}$$

For a test input, the model output is given by

$$f(k,l) = \kappa_0(b_1 + b_2 + a_{11}) + (\kappa_1 - \kappa_0)a_{k1} + (\kappa_1 - \kappa_0)a_{1l} = (\kappa_1 - \kappa_0)a_{k1} + (\kappa_1 - \kappa_0)a_{1l} =: f_1(k) + f_2(l). \tag{17}$$

This implies that

$$f_1(k) = \frac{\kappa_1 - \kappa_0}{\kappa_2 - \kappa_1}v_1(k), \quad f_2(l) = \frac{\kappa_1 - \kappa_0}{\kappa_2 - \kappa_1}v_2(l). \tag{18}$$

### D.1.2 DETAILED SIMULATIONS

Fig. 8b depicts the inferred values for kernels with various values for $\hat{\kappa}_1$, both for the training sets analyzed in Fig. 4 and two additional dispersed datasets (Fig. 8a). Lower $\hat{\kappa}_1$ consistently yields more distorted values.

We additionally analyze ReLU networks trained in the rich and lazy regime. The rich networks are equal in performance for extrapolation, worse for asymmetric extrapolation, and better for interpolation (Fig. 9a). They are generally well explained by an additive model, except for the asymmetric extrapolation (Fig. 9b). Fig. 9c confirms that the ReLU networks indeed change their representation in the rich regime: at initialization, the similarity between different trials is approximately clustered by whether those trials are distinct, overlapping, or identical. After lazy training, this remains unchanged, whereas after rich training, the similarities look entirely different. Finally, the inferred values look similar on extrapolation, but do not exhibit a memorization leak on interpolation (Fig. 9d). On asymmetric extrapolation, they look qualitatively different (though note that this network is also not additive). For the different dispersed tasks, the values inferred by the rich network largely follow the same patterns as those inferred by the lazy network, but also deviate occasionally.

Finally, the convolutional networks generally improve their generalization with increasing distance between the digits (Fig. 8a). All networks are extremely well explained by an additive model

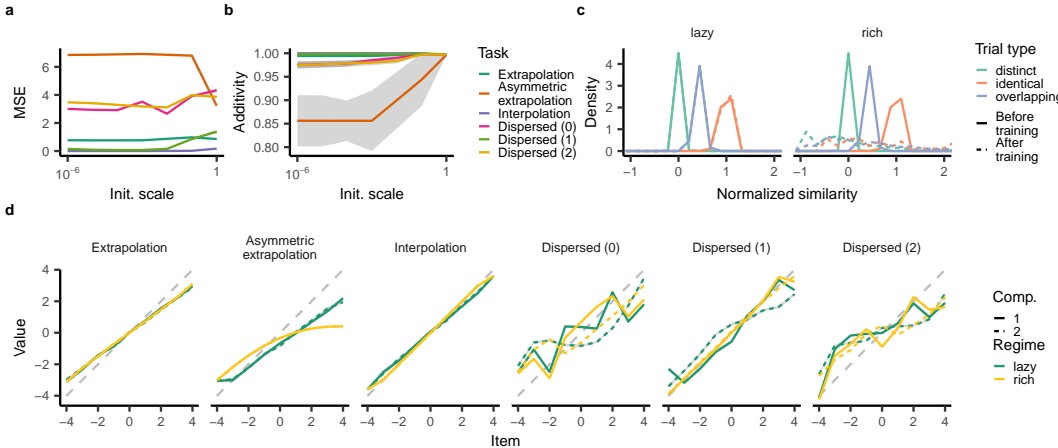

Figure 9: Behavior of rich neural networks trained on symbolic addition. **a**, **b**, Generalization (a) and additivity (b) as a function of initialization scale. **c**, Similarity between distinct, identical, and overlapping trials, before and after training in the lazy or rich regime. **d**, Inferred values in the lazy and rich regime.

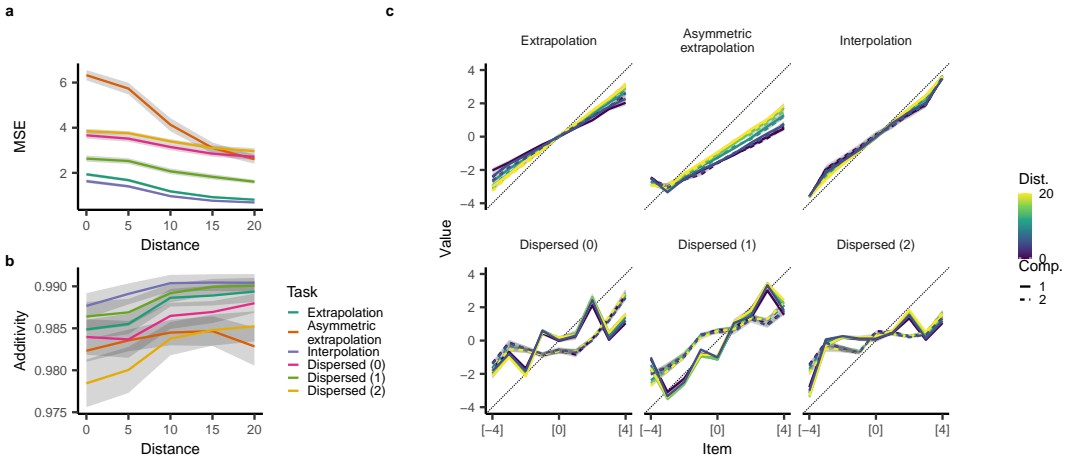

Figure 10: Behavior of convolutional networks trained on symbolic addition. **a**, **b**, Generalization (a) and additivity (b) as a function of distance. **c**, Inferred values for different distances.

($R^2 > 0.975$, Fig. 8b). Finally, the networks' inferred values look strikingly similar to those inferred by the kernel model (Fig. 8b) and their distortion tends to be higher for lower distances (i.e. a more conjunctive representation).

## D.2 CONTEXT-DEPENDENT DECISION-MAKING

### D.2.1 GENERAL TASK DEFINITION

We generally consider inputs with three components, $(\mathrm{co}, \mathrm{feat}_1, \mathrm{feat}_2)$. We assume that $\mathrm{co} \in C_1 \cup C_2$, where $C_1$ is the set of possible contexts under which $\mathrm{feat}_1$ is relevant and $C_2$ is the set of possible contexts under which $\mathrm{feat}_2$ is relevant. We further assume that there are decision functions $d_1(\mathrm{feat}_1), d_2(\mathrm{feat}_2) \in \mathbb{R}$. (For example, in the example in the main text, these function map three features to the first category (i.e. $y = -1$) and three features to the second category (i.e. $y = 1$).) The target is then given by

$$y(\mathrm{co}, \mathrm{feat}_1, \mathrm{feat}_2) = \begin{cases} d_1(\mathrm{feat}_1) & \text{if } \mathrm{co} \in C_1, \\ d_2(\mathrm{feat}_2) & \text{if } \mathrm{co} \in C_2. \end{cases} \tag{19}$$

Note that in the main text, we consider $C_1 = \{1\}$, $C_2 = \{2\}$, and $\text{feat}_1, \text{feat}_2 \in \{1, \dots, 6\}$ where

$$d_c(\text{feat}_c) = \begin{cases} 1 & \text{if } \text{feat}_c <= 3, \\ -1 & \text{else.} \end{cases} \tag{20}$$

### D.2.2 NOVEL STIMULUS COMPOSITIONS ARE CONJUNCTION-WISE ADDITIVE

If the test set consists in novel combinations of stimuli, this is a conjunction-wise additive computation. Namely, suppose that for all test inputs $(\text{co}, \text{feat}_1, \text{feat}_2)$, the two features have never been observed in conjunction, but both $(\text{co}, \text{feat}_1)$ and $(\text{co}, \text{feat}_2)$ have been. (This includes the case considered in the main text.) In this case, we can define functions $f_{12}$ and $f_{13}$ to implement the appropriate mapping:

$$f_{12}(\text{co}, \text{feat}_1) := \begin{cases} d_1(\text{feat}_1) & \text{if } \text{co} \in C_1, \\ 0 & \text{if } \text{co} \in C_2, \end{cases} \quad f_{13}(\text{co}, \text{feat}_2) := \begin{cases} 0 & \text{if } \text{co} \in C_1, \\ d_2(\text{feat}_1) & \text{if } \text{co} \in C_2, \end{cases} \tag{21}$$

$$f(\text{co}, \text{feat}_1, \text{feat}_2) = f_{12}(\text{co}, \text{feat}_1) + f_{13}(\text{co}, \text{feat}_2). \tag{22}$$

### D.2.3 NOVEL RULE COMPOSITIONS ARE NOT CONJUNCTION-WISE ADDITIVE

We could also imagine an alternative generalization rule, which we call **CDM (rule composition)**. In that case, there are multiple components indicating the same context: $C_1 = \{1, 2\}$ and $C_2 = \{3, 4\}$. We then leave out certain features with certain contexts. For example, $\text{feat}_1, \text{feat}_2 \in \{1, \dots, 6\}$, with the decision function defined in Eq. 20. Suppose we had never seen $\text{feat}_1, \text{feat}_2 \in \{3, 6\}$ in conjunction with $\text{co} \in \{2, 4\}$. In principle, if the model understood that $\text{co} = 1, 2$ (and $\text{co} = 3, 4$ resp.) signify the same context (i.e. learned to abstract the context from the context cue), it could generalize successfully as it had observed these features in conjunction with $\text{co} = 1, 3$. However, the conjunction-wise additive mapping depends on having observed each context in conjunction with each feature and this task is therefore non-additive.

### D.2.4 COEFFICIENT GROUPS

In Fig. 5, we grouped the inferred coefficients into categories. We here explain these categories:

- Right conj.: This is the correct conjunction the model should use to solve the task, i.e. between $\text{co} = 1$ and $\text{feat}_1$ and between $\text{co} = 2$ and $\text{feat}_2$.

- Wrong conj.: This is the incorrect conjunction between context and feature, i.e. between $\text{co} = 1$ and $\text{feat}_2$ and between $\text{co} = 1$ and $\text{feat}_2$.

- Sensory feat.: This is any conjunction involving sensory features, i.e. $\text{feat}_1$, $\text{feat}_2$, and $(\text{feat}_1, \text{feat}_2)$.

- Context only: This is the component co by itself.

- Memorization: This is the full conjunction of all three components $(\text{co}, \text{feat}_1, \text{feat}_2)$.

We then compute the average absolute magnitude within each of these groups in order to determine their overall relevance to model behavior.

### D.2.5 DETAILED SIMULATIONS

We find that rich networks generalize consistently on *CDM-1* and CDM-2 but not *CDM-3* (Fig. 11a). They are also perfectly conjunction-wise additive (Fig. 11b) and fail due to a context shortcut (Fig. 11c).

Generally, convolutional networks are also well explained by a conjunction-wise additive computation, though their additivity decreases for decreasing distance on *CDM-3* (Fig. 11c). We find that this coincides with a slightly lower magnitude associated with the context coefficient (consistent with the accuracy of the network increasing from below chance to chance level, Fig. 5d).

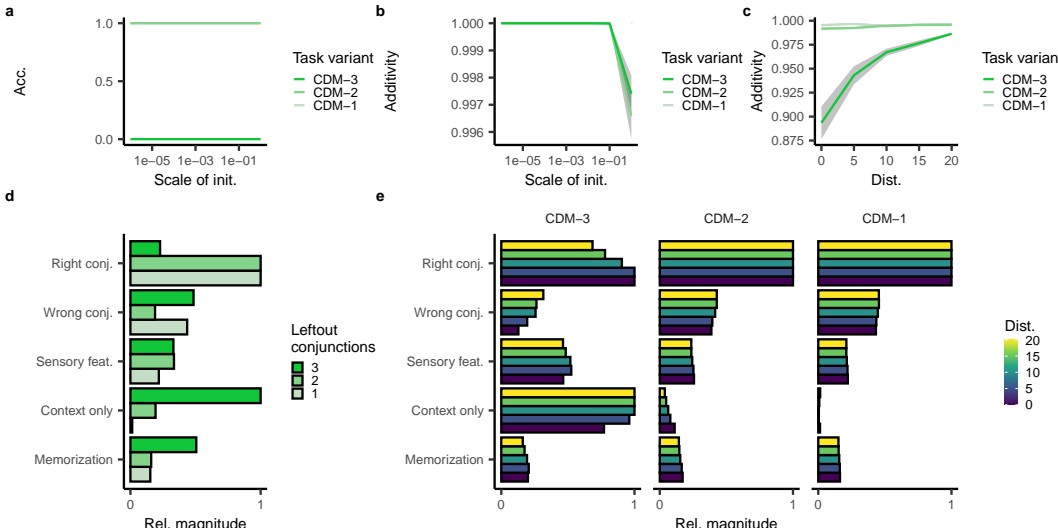

Figure 11: Behavior of rich and convolutional networks on context-dependent decision-making. **a**, The accuracy of rich networks is consistently either 1 (for CDM-2 and CDM-1) or 0 (for CDM-3). **b**, **c**, Additivity for rich networks (b) and convolutional networks (c). **d**, Magnitude of inferred values for rich networks. They also fail on CDM-3 due to a context shortcut. **e**, Magnitude of inferred values for convolutional networks with different distances.

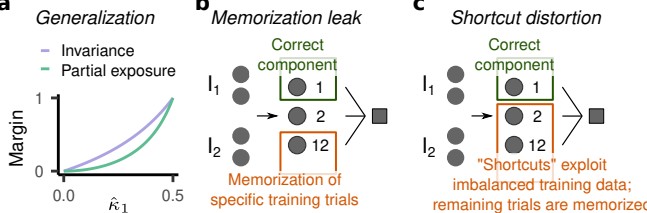

Figure 12: Two important failures for additive compositionality. **a**, Margin of generalization for component invariance and partial exposure, as a function of $\hat{\kappa}_1$. **b**, **c**, Schematics explaining the memorization leak (b) and the shortcut distortion (c).

### D.3 INVARIANCE

We analytically compute the kernel models' test set prediction on the invariance task. The training set is given by $\{(-1, -1), (-1, 1)\}$ and its kernel is therefore

$$K = \begin{pmatrix} \kappa_2 & \kappa_1 \\ \kappa_1 & \kappa_2 \end{pmatrix}, \tag{23}$$

where $\kappa_2$ is the similarity between identical trials and $\kappa_1$ is the similarity between overlapping trials. Hence, the dual coefficients are given by

$$a = K^{-1} \begin{pmatrix} 1 \\ -1 \end{pmatrix} = \frac{1}{\kappa_2^2 - \kappa_1^2} \begin{pmatrix} \kappa_2 & -\kappa_1 \\ -\kappa_1 & \kappa_2 \end{pmatrix} \begin{pmatrix} 1 \\ -1 \end{pmatrix} = \frac{1}{\kappa_2^2 - \kappa_1^2} \begin{pmatrix} \kappa_2 + \kappa_1 \\ -(\kappa_2 + \kappa_1) \end{pmatrix}. \tag{24}$$

The test set is given by $\{(1, -1), (1, 1)\}$ and its kernel with respect to the training set is therefore

$$\tilde{K} = \begin{pmatrix} \kappa_1 & \kappa_0 \\ \kappa_0 & \kappa_1 \end{pmatrix}, \tag{25}$$

where $\kappa_0$ is the similarity between distinct trials. Hence the test set predictions are given by

$$\hat{y} = \tilde{K}a = \frac{1}{\kappa_2^2 - \kappa_1^2} \begin{pmatrix} (\kappa_2 + \kappa_1)(\kappa_1 - \kappa_0) \\ -(\kappa_2 + \kappa_1)(\kappa_1 - \kappa_0). \end{pmatrix} \tag{26}$$

As the ground truth labels are $y = \{1, -1\}$, the margin $m = y\hat{y}$ is identical for both test set points:

$$m = \frac{(\kappa_2 + \kappa_1)(\kappa_1 - \kappa_0)}{\kappa_2^2 - \kappa_1^2} = \frac{\kappa_1 - \kappa_0}{\kappa_2 - \kappa_1} = \frac{(\kappa_2 - \kappa_0)\hat{\kappa}_1}{(\kappa_2 - \kappa_0) - (\kappa_1 - \kappa_0)} = \frac{\hat{\kappa}_1}{1 - \hat{\kappa}_1}. \tag{27}$$

### D.4 PARTIAL EXPOSURE

For partial exposure, the training set is given by $\{(-1, -1), (-1, 1), (1, -1)\}$ and its kernel is therefore

$$K = \begin{pmatrix} \kappa_2 & \kappa_1 & \kappa_1 \\ \kappa_1 & \kappa_2 & \kappa_0 \\ \kappa_1 & \kappa_0 & \kappa_2 \end{pmatrix}. \tag{28}$$

The test set is given by $\{(1, 1)\}$ and the test set kernel is therefore

$$\tilde{K} = \begin{pmatrix} \kappa_0 & \kappa_1 & \kappa_1 \end{pmatrix} \tag{29}$$

The margin is therefore given by

$$m = y\hat{y} = -\hat{y} = -\tilde{K}K^{-1} \begin{pmatrix} 1 \\ -1 \\ 1 \end{pmatrix}. \tag{30}$$

We solve this equation for the special case where $\kappa_0 = 0$ and $\kappa_1 = 1$ using Mathematica and find that

$$m = \frac{2\kappa_1^2}{1 - 2\kappa_1^2}. \tag{31}$$

### D.5 OTHER MATHEMATICAL OPERATIONS

We could consider mathematical operations other than addition as well, considering unobserved assigned values $v_1[I_1]$ and $v_2[I_2]$ together with some composition function $C(v_1[I_1], v_2[I_2])$. This task will only be additive if the composition function is additive (e.g. if it is subtraction). If it is, e.g. multiplication, division, or exponentiation, the task will be non-additive.

### D.6 LOGICAL OPERATIONS

In this task, inputs with two components are presented. Each component $I_c$ has an unobserved truth value $T[I_c]$ associated with it and the target is some logical operation over these two truth values, for example *AND*: $T[I_1] \wedge T[I_2]$. After inferring the truth value of each component, the model could generalize towards novel item combinations. As long as the logical operation is additive (e.g. *AND*, *OR*, *NEITHER*, . . . ), this is an additive task. If the logical operation is non-additive (e.g. *XOR*), this would be a non-additive task. Indeed, this case would correspond to the transitive equivalence task.

## D.7 TRANSITIVE ORDERING

Transitive ordering (TO) is a popular task in cognitive science (often called transitive inference, McGonigle & Chalmers 1977). Here the subject is presented with two items $I_1$, $I_2$ drawn from an unobserved hierarchy $>$. It should then categorize whether $I_1 > I_2$ or $I_2 > I_1$. Crucially, this task can be solved by assigning a rank $r(I_c)$ to each item and computing the response as $f(I) = r(I_1) - r(I_2)$ (**?**). It is therefore additive, in contrast to transitive equivalence. This is also the case if we assume that there are multiple such hierarchies (e.g. $a_1 > \ldots, a_5$ and $b_1 > \ldots, b_5$). In this case, the model would generalize to comparisons between these different hierarchies as well (**cross-list TO**).

