# OpenReview forum: "The impact of task structure, representational geometry, and learning mechanism on compositional generalization"
_ICLR.cc/2024/Workshop/Re-Align — ICLR 2024 Workshop Re-Align Poster_

### Official Review · Reviewer_6gmQ · 2024-02-19
**A solid paper, could be made more readable**

**Rating:** 2
**Fit:** 2
**Confidence:** 2

**Workshop Review:**

The authors carried out a taxonomy of compositionality tasks from the perspective of being learnable by kernel methods or rich networks. They found that kernel methods can only represent conjunction-wise additive tasks, even with two failure modes called memorization leak and shortcut learning. By contrast, networks in the rich regime have more expressive power and generalize better than kernel methods.

The paper is generally well written and well structured. The motivation of the paper can be made clearer - there are five contribution points, making it a bit hard to tell which is the major contribution. I feel many important statements should be stated more clearly:

(1) Below definition 3.1, it is not clear to me why "Note that if r(x) arises from a deep neural network with onehot inputs and random weights, it is compositionally structured in expectation.".

(2) Section 5.2, how are memorization leak and shortcut distortion related to literature? Are these only failure modes for kernel methods, or are they also problematic for rich networks?

(3) At the top of page 18, " In particular, when reporting results on rich networks (without further specification), we assume 10−6. When reporting results on lazy network, we assume 10−6." Why do lazy and rich networks have the same scale?

**Reason For Not Giving Higher Score:**

The paper has several valuable points but they are scattered and not significant enough.

**Reason For Not Giving Lower Score:**

The writing of the paper is very solid and their results are well supported by experiments.

**Reviewer Domain:**

machine learning

---

> ### Author Response · Authors · 2024-05-10
>
> Thank you very much for your helpful review! We are encouraged that you think our paper makes valuable points and is generally well written. We also appreciate your feedback on clarity and readability, which we will make sure to take into account for future revisions of this paper.
>
> "Below definition 3.1, it is not clear to me why "Note that if r(x) arises from a deep neural network with onehot inputs and random weights, it is compositionally structured in expectation."."
>
> We now clarify this in the updated draft: "Note that if $r(x)$ arises from a random deep neural network with inputs concatenating onehot vectors for each component, it is compositionally structured in expectation. This is because this input is compositionally structured (as the similarity between two inputs is equal to their number of overlaps) and the output kernel of densely connected neural networks with random weights only depends on their input kernel (i.e. for a network $\phi$, if $K(x_1,x_1')=K(x_2,x_2')$, then $K(\phi(x_1),\phi(x_1'))=K(\phi(x_2),\phi(x_2'))$)."
>
> "At the top of page 18, " In particular, when reporting results on rich networks (without further specification), we assume 10−6. When reporting results on lazy network, we assume 10−6." Why do lazy and rich networks have the same scale?"
>
> This was a typo -- the scale of the lazy network should be $1$. We have now corrected this.

---

### Official Review · Reviewer_WV5X · 2024-02-21

**Rating:** 2
**Fit:** 3
**Confidence:** 2

**Workshop Review:**

**Summary**

The paper presents a significant step towards formalizing the relationship between model structure and compositional behaviors. The authors delineate the range of compositional computations that can be implemented by kernel models, identifying a crucial distinction between additive and non-additive tasks. They highlight failure modes for additive tasks and demonstrate how rich neural networks can navigate beyond the limitations of conjunction-wise additivity. The theoretical framework is empirically validated through analyses of deep neural network behaviors, positioning the work as a valuable contribution to understanding model generalization.

**Strengths**
- The paper introduces a novel framework for understanding the compositional capabilities of kernel models, providing a fresh perspective on the differentiation between additive and non-additive tasks.
- By highlighting specific failure modes in kernel models, the authors offer insights into why models may fail at compositional generalization, contributing to the broader discourse on improving model robustness and reliability.
- The validation of their theoretical insights through empirical analysis strengthens the paper's credibility, showcasing the practical relevance of their findings.

**Weaknesses**
- The primary limitation noted is the focus on smaller models and toy domains for analytical tractability. This raises questions about the scalability of the findings and their applicability to larger, more complex neural networks and real-world tasks.

**Questions for Authors**
- How do the authors envision the application of their findings to complex compositional generalization problems beyond the controlled experimental settings presented in the paper?
- The paper could benefit from clearer explanations, particularly concerning the evaluation intentions of CDM variants in Figure 1d.

**Reason For Not Giving Higher Score:**

N/A

**Reason For Not Giving Lower Score:**

N/A

**Reviewer Domain:**

machine learning

---

> ### Author Response · Authors · 2024-05-10
>
> Thank you very much for your helpful review!
>
> "How do the authors envision the application of their findings to complex compositional generalization problems beyond the controlled experimental settings presented in the paper?"
>
> This is a direction we are very interested in exploring. Here we wanted to focus on smaller-scale tasks, as, in our opinion, they provide useful "building blocks" for larger-scale tasks. In particular, more complex compositional generalization problems could be decomposed into these simple building blocks and our theory, in this way, may be able to speak to such more complex settings as well (either empirically or through an extension of our theory).
>
> "The paper could benefit from clearer explanations, particularly concerning the evaluation intentions of CDM variants in Figure 1d."
>
> We have updated the caption to Figure 1 clarify this point.

---

### Official Review · Reviewer_2jZG · 2024-02-26
**A good piece of work and worth addition to the workshop**

**Rating:** 3
**Fit:** 3
**Confidence:** 3

**Workshop Review:**

## Clarity
The paper is well written and clear. Figures are also helpful. The captions could be more descriptive, particularly for Figure 1 which is relied on heavily to explain the context task. The absence of a clear explanation there in the text or caption means I spent a bit of time trying to interpret the image. For example Figure1e does not say why the different components of the $2\times2$ matrices have different magnitudes. Figure 5 is an example where the caption is more helpful. Some notation or terminology could be introduce but there were no cases where I could not figure out what was intended. For example, where you say "i.e. if O(I, I ′ ) = O(J, J ′ ), then K(I, I ′ ) = K(J, J ′ )" - is J just some other task? In that case are I,I',J and J' four different tasks and we are looking for common shared structure between (I and I') and (J and J')? The sentence before this notation is explanatory and so I could rely on that. Another example is the overlap variable: $\kappa_i$ is taken for granted where it could just be introduced more fully. A final is: "...end-to-end training of these architectures often does not result in the desired modular specialization. Indeed, even standard network architectures can develop specialized modules...", these two sentences seem to be contradictory. But I get the overall point and again, in general the paper is well written. The distinction between conjunction-wise and component-wise additivity is an example of the paper introducing a concept clearly.

## Correctness
Again I think overall this work is of a high quality and the results and conclusion are correct. I have also gone through the appendix briefly too. I think the characterization in the introduction that this work is more general than prior work in terms  of task is slightly premature. I acknowledge that Propositions 4.1 and 4.2 make no constraints on the kind of structure and so it is very general. But the idea of combining pieces of structure to assess generalization can be found in Jarvis et. al. (2023) where they also use a space of tasks, rather than just one (they do limit the kinds of structure unlike the two given propositions in this work). On the other end Section 5 onwards considers binary inputs and so appears similar in task scope to Abbe et. al. and Lippl et. al. This does not detract from the work for me but I think being clear about these things is important. Finally, there is another more recent theoretical work by Schug et. al. which the authors might find interesting on transitive generalization. My only other concern on the correctness is the interpretation of Propositions 4.1 and 4.2. The authors state that they present an impossibility theorem due to the fact that conjunctive factors will appear which cannot then be used for generalization. But as far as I can tell, the proof or proposition statement does not actually guarantee that these functions are going to have a material impact on the labels. Perhaps I am not tying this in enough with the lazy regime, but it seems to me you could have this conjunctive part of the function but the magnitude of its output be small. There is probably a no free lunch-style argument here but I am just curious how we could be certain that this portion of the function decomposition is detrimental. A final question, for the statement "While many kernel models can learn arbitrary functions (Cybenko, 1989), our result shows that their compositional generalization is constrained to an additive
computation." - for the lazy regime, what is the alternative since all it has to work with is the random features and a linear readout? I hope these questions and points help the authors with subsequent submissions and should not be interpreted as overly critical.

## Novelty and Interest
I think this is an important step in a recent and relevant line of work. This work also has some interesting an unintuitive conclusions. For example the fact that depth leads to more conjunctivity and worse generalization. Additionally, the relation on the difficulty of tasks in Section 5.3 was nice.

- Emmanuel Abbe, Samy Bengio, Aryo Lotfi, and Kevin Rizk. Generalization on the Unseen, Logic
Reasoning and Degree Curriculum. June 2023. URL https://openreview.net/forum?
id=3dqwXb1te4.
- Devon Jarvis, Richard Klein, Benjamin Rosman, and Andrew M. Saxe. On The Specialization of
Neural Modules. In The Eleventh International Conference on Learning Representations, 2023.
URL https://openreview.net/forum?id=Fh97BDaR6I.
- Samuel Lippl, Kenneth Kay, Greg Jensen, Vincent P. Ferrera, and L. F. Abbott. A mathematical theory
of relational generalization in transitive inference, August 2023. URL https://www.biorxiv.
org/content/10.1101/2023.08.22.554287v1. Pages: 2023.08.22.554287 Section:
New Results.
- Schug, Simon, et al. "Discovering modular solutions that generalize compositionally." arXiv preprint arXiv:2312.15001 (2023).

**Reason For Not Giving Higher Score:**

I am giving a score of 3 but this is subject to the belief that some of my suggestions will be included in the updated draft. Otherwise a score of 2 seems appropriate.

**Reason For Not Giving Lower Score:**

The work is overall very nice and for a workshop I believe could spark some very interesting conversation. The topic is also quite popular at the moment and so I believe relevant for a wider audience and quite timely.

**Reviewer Domain:**

machine learning

---

> ### Author Response · Authors · 2024-05-10
>
> Thank you very much for your helpful and encouraging review! We have updated our draft in accordance with your comments. In particular, we have clarified the caption for figure 1 and clarified the parts of the text you had highlighted.
>
> "I think the characterization in the introduction that this work is more general than prior work in terms of task is slightly premature. I acknowledge that Propositions 4.1 and 4.2 make no constraints on the kind of structure and so it is very general. But the idea of combining pieces of structure to assess generalization can be found in Jarvis et. al. (2023) where they also use a space of tasks, rather than just one (they do limit the kinds of structure unlike the two given propositions in this work). On the other end Section 5 onwards considers binary inputs and so appears similar in task scope to Abbe et. al. and Lippl et. al. This does not detract from the work for me but I think being clear about these things is important."
>
> Thank you for highlighting this; we now discuss in more detail in the related work section how our work differs from previous work. As you note, compared to Lippl et al. and Jarvis et al., we don't assume a specific input-output relation in Section 4. We see this as an important step forward, but now note in the related work section that this applies specifically to Section 4.
>
> "Finally, there is another more recent theoretical work by Schug et. al. which the authors might find interesting on transitive generalization."
>
> Thank you for bringing this to our attention, it's indeed very relevant.
>
> "My only other concern on the correctness is the interpretation of Propositions 4.1 and 4.2. The authors state that they present an impossibility theorem due to the fact that conjunctive factors will appear which cannot then be used for generalization. But as far as I can tell, the proof or proposition statement does not actually guarantee that these functions are going to have a material impact on the labels. Perhaps I am not tying this in enough with the lazy regime, but it seems to me you could have this conjunctive part of the function but the magnitude of its output be small. There is probably a no free lunch-style argument here but I am just curious how we could be certain that this portion of the function decomposition is detrimental."
>
> The impossibility theorem concerns the fact that these conjunctive factors cannot appear with respect to the test data (i.e. any sub-conjunction that has not been seen during training cannot be assigned a particular value). This constrains the kinds of functions the model can learn on the test set. Importantly, this is independent from whether those conjunctions are used to learn the training set. In practice, we observe that norm minimization causes the model to assign substantive magnitude to the conjunctive factors, giving rise to a memorization leak (see Section 5.2). However, this is not what the impossibility theorem is about. We have tried to clarify this further in the text.
>
> "A final question, for the statement "While many kernel models can learn arbitrary functions (Cybenko, 1989), our result shows that their compositional generalization is constrained to an additive computation." - for the lazy regime, what is the alternative since all it has to work with is the random features and a linear readout?"
>
> By additive computation, we mean summing up a value associated with each component. This is a subset of the functions that can be implemented by a linear readout of the random features (which could also associated specific values with each conjunction, as they do on the training set). We now clarify this in the text.
>
> "I hope these questions and points help the authors with subsequent submissions and should not be interpreted as overly critical."
>
> They are indeed very helpful to us as we are preparing a more substantive revision. Thank you very much for your comments and questions!

---

### Decision · Program_Chairs · 2024-03-02

Accept (Poster)